# A Potential Association between Abdominal Obesity and the Efficacy of Humoral Immunity Induced by COVID-19 and by the AZD1222, Convidecia, BNT162b2, Sputnik V, and CoronaVac Vaccines

**DOI:** 10.3390/vaccines12010088

**Published:** 2024-01-15

**Authors:** Javier Angeles-Martinez, Irma Eloisa Monroy-Muñoz, José Esteban Muñoz-Medina, Larissa Fernandes-Matano, Ángel Gustavo Salas-Lais, Ma De Los Ángeles Hernández-Cueto, Eyerahi Bravo-Flores, Moisés León-Juárez, Clara Esperanza Santacruz-Tinoco, Daniel Montes-Herrera

**Affiliations:** 1Central Epidemiology Laboratory, Mexican Social Security Institute, Mexico City 02990, Mexico; angel.salas@imss.gob.mx (Á.G.S.-L.); maria.hernandezcu@imss.gob.mx (M.D.L.Á.H.-C.); 2Reproductive and Perinatal Health Research Department, National Institute of Perinatology, Mexico City 11000, Mexico; irmae4901@gmail.com; 3Quality of Supplies and Specialized Laboratories Coordination, Mexican Social Security Institute, Mexico City 07760, Mexico; jose.munozm@imss.gob.mx (J.E.M.-M.); larissa.fernandes@imss.gob.mx (L.F.-M.); 4Southern Medical Line, Mexico City 11000, Mexico; eyerahiqfb@yahoo.com.mx; 5Immunobiochemistry Department, National Institute of Perinatology, Mexico City 11000, Mexico; moisesleoninper@gmail.com; 6Specialized Laboratories Division, Mexican Social Security Institute, Mexico City 07760, Mexico; clara.santacruz@imss.gob.mx

**Keywords:** abdominal obesity, COVID-19, vaccines, humoral immune response

## Abstract

Abdominal obesity is highly prevalent in Mexico and has a poor prognosis in terms of the severity of coronavirus disease (COVID-19) and low levels of antibodies induced by infection and vaccination. We evaluated the humoral immune response induced by COVID-19 and five different vaccination schedules in Mexican individuals with abdominal obesity and the effects of other variables. This prospective longitudinal cohort study included 2084 samples from 389 participants. The levels of anti-S1/S2 and anti-RBD IgG antibodies were measured at various time points after vaccination. A high prevalence of hospitalization and oxygen use was observed in individuals with abdominal obesity (AO) who had COVID-19 before vaccination; however, they also had high levels of anti-S1/S2 and anti-RBD-neutralizing IgG antibodies. The same was true for vaccination-induced antibody levels. However, their longevity was low. Interestingly, we did not observe significant differences in vaccine reactogenicity between abdominally obese and abdominally non-obese groups. Finally, individuals with a higher body mass index, older age, and previous COVID-19 had higher levels of antibodies induced by COVID-19 and vaccination. Therefore, it is important to evaluate other immunological and inflammatory factors to better understand the pathogenesis of COVID-19 in the presence of risk factors and to propose effective vaccination schedules for vulnerable populations.

## 1. Introduction

Coronavirus disease (COVID-19) is caused by the severe acute respiratory syndrome-coronavirus-2 (SARS-CoV-2), which has spread worldwide since the first cases were reported in 2019, becoming a pandemic in 2020 [1]. Various comorbidities have been implicated as risk factors for deterioration or poor outcomes in patients with COVID-19. Among the comorbidities, obesity is one of the most significant [2], although the underlying factors have not been fully elucidated.

Obesity has a high prevalence worldwide. In 2016, the Worldwide Health Organization (WHO) reported that 39% of adults were overweight and 13% were obese [3]. In Mexico, the situation was even more alarming; according to data from the National Health and Nutrition Survey (ENSANUT) 2018–19, the prevalence of overweight was 39.1%, obesity was 36.1%, and abdominal adiposity was 81.6% [4]. Having a high prevalence of individuals with obesity and overweight in Mexico makes the population more susceptible to complications from COVID-19. There are reports of a negative influence of obesity on the immune response against SARS-CoV-2, as people with a higher body mass index (BMI) have a higher risk of hospitalization, more severe symptoms of the disease, a higher risk of mechanical ventilation [5,6,7], and lower antibody levels induced by COVID-19 and vaccination [8,9].

However, BMI can be used only as an approximation of the degree of adiposity [3] because it does not consider body fat distribution, especially visceral fat. Visceral fat is a risk factor for several cardiometabolic diseases and is associated with a high mortality [10]. Waist/hip ratio is associated with a higher risk of death, cardiovascular disease, and diabetes than BMI. Waist circumference (WC) is strongly associated with the absolute amount of abdominal and visceral fat [11]. Abdominal adipose tissue has been reported to express several proinflammatory mediators that can generate a systemic inflammatory state and compromise lung function [12,13]. Likewise, central obesity is associated with an increased risk of developing severe COVID-19, a high risk of hospital admission, and with long hospital stays [14,15,16]. Excess visceral fat ≥ 128.5 cm^2^ is independently associated with the severity of COVID-19 [17]. Graziano et al. showed that visceral adiposity is a more sensitive parameter than BMI for predicting negative COVID-19 outcomes [18], as well as in other studies in which people with abdominal obesity (AO) had low levels of vaccination-induced antibodies in the Chinese population [19] and health workers from Italy [20,21]. In Mexico, different vaccines against COVID-19 were administered to the general population, allowing the evaluation of the effect of AO on the production of antibodies induced by the infection or by the application of the vaccines and their reactogenicity (physical manifestations of the inflammatory response to vaccination: fever, tiredness, headache, muscle pain, shaking chills, diarrhea, and local reaction).

Therefore, our main objective was to evaluate the impact of AO on the IgG antibody response induced by SARS-CoV-2 infection or from the application of any of the five vaccination schemes in Mexican individuals. Second, we analyzed the role of AO in symptomatology and reactogenicity generated by the application of vaccines. Finally, we evaluated the effect of other factors, such as documenting prior COVID-19, BMI, age, sex, comorbidities, and type of vaccine administered, on antibody levels.

## 2. Materials and Methods

### 2.1. Ethics Statements

This study was conducted according to the guidelines of the Declaration of Helsinki and approved by the Ethics and Scientific Committee of the National Scientific Research Commission of the IMSS (Mexican Institute of Social Security). This study is part of the project “Seroprevalence and neutralizing activity of SARS-CoV-2 IgG antibody seroprevalence sera in Mexico” approved on 13 March 2020 with registration number R-785-2020-60. The study participants were selected based on voluntary application, and informed consent was obtained from all subjects. For the monitoring of the participants and analysis of the samples, an amendment to the project was requested, which was approved on 10 August 2022 with registration number R-2022-785-037.

### 2.2. Study Design and Population

A prospective, longitudinal cohort study was conducted to evaluate the immunological status of the Mexican population before and after the application of any of the five vaccination schemes against SARS-CoV-2. Participants were recruited between 14 May and 27 July 2021, and were monitored during hospital follow up for 12 months after the first dose of the vaccine was administered. A total of 389 participants were included, and the number of participants recruited for each vaccination scheme was as follows: 100 from AZD1222, 84 from Convidecia, 70 from BNT162b2, 80 from Sputnik V, and 55 from CoronaVac. 

All participants received two inoculations of the COVID-19 vaccine separated by 21 d (BNT162B2), 90 d (AZD1222, Sputnik V, and CoronaVac), and 8 months (Convidecia participants received Moderna as a booster dose). Blood samples were obtained from all participants on the day of the first inoculation of their vaccine (sample 1) and later at 21, 90, 180, 270, and 365 d (samples 2, 3, 4, 5, and 6, respectively). A booster (third dose) was applied to the general population in January 2022; of the total number of participants, only 84.8% (330 participants) received this vaccine dose. The main vaccine administered for the booster was AZD1222 (57.9%), followed by Sputnik V (22.7%) and Moderna (16.7%). Self-reported adverse events or reactogenicity were collected using a questionnaire on day 21 after the first dose (sample 2) and on day 90 (sample 3) after the application of the second dose of each vaccination schedule, except for the vaccination schedule, Convidecia, which was at 270 days (Moderna vaccine booster). The inclusion criteria were as follows: participants who attended the application of their first vaccine dose and who agreed to sign an informed consent. Participants were excluded if they were missing serological results at more than two of the six time points studied, if they had a vaccination dose prior to the date of inclusion, or if they developed any respiratory symptoms at the time of inclusion.

### 2.3. Demographic Variables

The data obtained included baseline demographic information, such as sex (women and men); age (years); medication use; the presence of comorbidities, such as hypertension (HTA), type 2 diabetes (T2D), autoimmune disease (AD), other respiratory diseases (ORD), or heart disease (HD); and history of smoking, alcohol use, or drug use. Participants with COVID-19 prior to their vaccination schedule were defined as those who reported a previous infection (positive results for real-time polymerase chain reaction, antigen and antibody tests, and clinical symptomatology). The data were processed and analyzed without any personal identifiers to maintain participant confidentiality.

### 2.4. Anthropometric Measures

Measurements included weight (kg) and height (m), as reported by the patient. BMI was then calculated using the formula BMI = weight (kg)/height^2^ (m) and participants were classified according to BMI as normal weight (BMI < 25 kg/m^2^, overweight BMI 25–29, 9 kg/m^2^, or obese BMI > 30 kg/m^2^. The participants’ WC was measured in meters (m) at the midpoint between the last rib and the iliac crest in the patient standing and exhaling; participants with a WC > 1.02 m in men and >0.88 m in women were defined as having abdominal obesity (AO+) and participants with a circumference below these values were defined as participants without abdominal obesity (AO−), according to the thresholds proposed by the National Institutes of Health (NIH).

### 2.5. Detection of Antibody Levels against SARS-CoV-2

The detection of IgG antibodies against the S1/S2 antigens of SARS-CoV-2 and the neutralizing activity of antibodies against the RBD antigen of SARS-CoV-2 in blood samples were analyzed as previously described [22].

### 2.6. Statistical Analysis

Baseline characteristics of AO− and AO+ participants were reported based on demographic and clinical characteristics of interest using counts and percentages to determine differences between both populations. GraphPad Prism version 8 software (GraphPad Software, San Diego, CA, USA) was used to analyze data and generate figures. Continuous variables were compared between the AO− and AO+ groups and each vaccine complex, using the non-parametric Mann–Whitney and Kruskal–Wallis U tests; nominal variables were analyzed using the chi-square test or Fisher’s exact test. Correlation was assessed by calculating non-parametric Spearman correlation with a two-tailed 95% confidence interval (CI). The SPSS statistical package (version 21.0; SPSS, Chicago, IL, USA) was used to generate a multiple linear regression analysis to determine the relationship between the anti-S1/S2 and anti-RBD-neutralizing IgG antibody levels and the variables.

## 3. Results

### 3.1. Characteristics of the Study Population

This analysis included 2084 samples from 389 participants (132 men and 257 women), of which 218 (56.04%) presented with AO. Of the five vaccination schemes included in the study, the BNT162b2 group presented the highest percentage of AO (65.7%) and the lowest percentage of Convidecia (46.4%). Interestingly, this group also had the highest median body weight (80 kg). The participants in the Convidecia, BNT162b2, and Sputnik V schemes in the AO+ group presented the highest median WC (1.03 m). The BNT162b2 AO+ group was the oldest group (54.5 y) in our study. The comorbidity with the highest prevalence in AO+ participants was hypertension, which was significant only in the AZD1222 group (*p* < 0.05); this group also had the lowest prevalence of autoimmune diseases (*p* < 0.05). The group with the AZD1222 vaccination scheme showed significant differences when comparing the seronegative, asymptomatic, and symptomatic groups between the AO+ and AO− groups (Table 1). 

### 3.2. Antibody Response Induced by COVID-19 in AO+ Individuals

In our study cohort, 21.9% of the participants had documented prior COVID-19, of whom 63.5% had AO. The prevalence of hospitalization and oxygen use was high in the AO+ group (Figure 1A). In addition, 26.2% of the total number of participants presented with COVID-19 after the start of their vaccination schedule, and 54.9% presented with AO. Finally, a decrease in the prevalence of hospitalization and the use of oxygen for infections was observed after the start of the vaccination schedule in both AO− and AO+ participants (Figure 1B). 

To assess the effect of AO on the IgG antibody response in a SARS-CoV-2 infection, we measured the levels of anti-S1/S2 IgG antibodies and anti-RBD-neutralizing antibodies in AO+ and AO− participants who documented previous COVID-19 (basal sample: day 0). The AO+ participants had higher levels of anti-S1/S2 and anti-RBD IgG-neutralizing antibodies than the AO− participants. Subsequently, we stratified the AO+ and AO− participants with prior COVID-19 as asymptomatic or symptomatic. We observed that asymptomatic AO+ participants had high levels of anti-S1/S2 and anti-RBD-neutralizing IgG antibodies, without a significant difference. In the symptomatic group, the AO+ participants had high levels of anti-S1/S2 and anti-RBD-neutralizing IgG antibodies (Table 2). We found a positive correlation between WC and the levels of anti-S1/S2 and anti-RBD-neutralizing IgG antibodies in participants with previous COVID-19. When the participants were stratified into asymptomatic and symptomatic groups, we observed a stronger correlation with the levels of anti-S1/S2 IgG antibodies in symptomatic participants than in asymptomatic participants.

Next, we evaluated whether AO influenced antibody production after the administration of any of the vaccination schemes in participants with or without prior COVID-19. We found that in the group of participants without prior COVID-19, AO+ participants had higher levels of anti-S1/S2 and anti-RBD-neutralizing IgG antibodies at the time points analyzed, with significant differences observed only at 21 d. The group with previous COVID-19 showed significant differences only in anti-S1/S2 IgG antibody levels at 270 and 365 d (Table 2).

### 3.3. Reactogenicity of the Different Vaccination Schedules Applied in the Population with AO

The vaccines administered to the participants were well-tolerated. The duration of any adverse reactions in our cohort was <3 d after the first and second doses, except for the second dose of AZD1222, in which a high frequency of participants without any reactions was found. The most frequent reaction after each dose was pain at the injection site (Figure 2). Muscle pain was the most frequent reaction to the AZD1222 vaccine in participants with AO after the first dose and fatigue after the second dose (Figure 2A). The analysis of the Convidecia, BNT162b2, Sputnik V and CoronaVac groups showed no significant differences (Figure 2B–E). 

### 3.4. Prior COVID-19, Abdominal Obesity, and Age Are Factors Independently Associated with Antibody Response

Multiple linear regression analyses were performed to identify risk factors associated with humoral immune responses at different time points. In our population, on day 0, we observed that abdominal obesity, age, and prior COVID-19 were factors associated (all *p* < 0.01) with the IgG antibody response against SARS-CoV-2 (Table 3). These factors explained 15% of the variance in anti-S1/S2 IgG antibody levels and 10.5% (BMI class instead of AO) of the variance in anti-RBD-neutralizing antibody levels [anti-S1/S2 IgG: F(3.389) = 9.516, *p* < 0.0001; IgG anti-RBD-neutralizing antibodies: F(3.389) = 6.282, *p* < 0.0001]. Among the different time points analyzed in this study, the main factor associated with anti-S1/S2 and anti-RBD-neutralizing IgG antibody responses was prior to COVID-19 (all *p* < 0.001). 

Interestingly, age had a positive association with the levels of anti-RBD-neutralizing antibodies induced by previous COVID-19 (day 0) and at day 90 but a negative association was observed at 270 and 365 d. An additional model in the general population on days 180 and 365 showed that vaccination schedules were a significant factor influencing the IgG antibody response (*p* < 0.014 and *p* = 0.002, respectively). In addition, at 270 d, sex was a significant factor in the anti-S1/S2 IgG antibody response and T2D for anti-RBD-neutralizing antibodies (*p* = 0.006 and *p* = 0.005, respectively). At 365 d, alcoholism was a significant factor (*p* = 0.025).

### 3.5. IgG antibody Response Induced by the Different Vaccination Schemes in AO+ Individuals

To evaluate the humoral immune response induced by the application of any vaccination scheme, we analyzed the levels of anti-S1/S2 IgG antibodies and the neutralizing activity of specific anti-RBD-neutralizing antibodies in AO− and AO+ individuals at different time points (21, 90, 180, 270, and 365 d) after the application of the first dose of the vaccine. The AO+ participants had high levels of anti-S1/S2 IgG antibodies at 21, 90, 180, 270, and 365 d. Regarding the neutralizing activity of anti-RBD-neutralizing antibodies, a similar phenomenon occurred at 21 and 90 d (Figure 3A). 

Stratifying by vaccination schedules, AZD1222 AO+ participants had significantly higher levels of anti-S1/S2 IgG antibodies than AO− at 21, 270, and 365 d. Similar to the above, AO+ participants had higher levels of neutralizing anti-RBD-neutralizing antibodies than AO− at 21 d (Figure 3B). In the Convidecia vaccine participants, we found that AO+ had higher levels of anti-RBD-neutralizing antibodies than AO− at 21 d (Figure 3C). The group administered BNT162b2 vaccine did not show any significant differences (Figure 3D). Sputnik V AO+ participants had high levels of anti-S1/S2 IgG antibodies at 90 d, and anti-RBD-neutralizing antibodies with neutralizing activity at 21 and 90 d (Figure 3E). With the CoronaVac vaccine, AO+ participants had high levels of IgG anti-S1/S2 antibodies at 21, 90, and 270 d, and the levels of anti-RBD-neutralizing antibodies were high at 21 and 90 d (Figure 3F).

Additionally, we evaluated the correlation between WC and levels of anti-S1/S2 and anti-RBD-neutralizing IgG antibodies. First, in the total population, regardless of the vaccination scheme, we found a discrete but significant correlation between WC and IgG antibody levels at 21, 90, 270, and 365 d as well as with the neutralization levels of anti-RBD-neutralizing antibodies only at 21 and 90 d (Figure 4A). The participants who received the AZD1222 vaccine showed a correlation at 21 and 270 d with the levels of IgG anti-S1/S2 antibodies and at 21 d with the levels of anti-RB- neutralizing antibodies (Figure 4B). The Convidecia group presented a slight correlation with the levels of IgG anti-S1/S2 antibodies at 21 and 365 d and with the levels of anti-RBD-neutralizing antibodies at 21 d (Figure 4C). The group with BNT162b2 did not show any significant differences (Figure 4D). The Sputnik V group showed a slight correlation with the levels of anti-RBD-neutralizing antibodies at 90 d (Figure 4E). Finally, the CoronaVac vaccine was correlated with the levels of IgG anti-S1/S2 antibodies at 21 and 90 d (Figure 4F).

In a complementary analysis to determine whether BMI had comparably strong associations as WC, we observed that the higher the BMI, the higher the levels of anti-S1/S2 and anti-RBD-neutralizing IgG antibodies induced by a previous infection in the general population. When analyzing the levels of antibodies induced by vaccination in the total population, we observed that they were only maintained at 21 and 90 d with anti-S1/S2 IgG antibodies. When stratified by the vaccination scheme, AZD122 was maintained on day 21, similarly to Convidecia. Sputnik V correlated at 21 d with IgG anti-S1/s2 and anti-RBD-neutralizing antibodies on days 21 and 90. BNT162b2 and CoronaVac vaccines did not show any significant differences (Appendix A).

### 3.6. Abdominal Obesity Influences the Seropositivity of Participants without COVID-19 Prior to the Application of Their Vaccination Scheme

The group of participants who did not document prior COVID-19 allowed us to determine the seropositivity rate of anti-S1/S2 and anti-RBD-neutralizing IgG antibodies after administration of the first, second, and third doses of each vaccination scheme analyzed in the AO− and AO+ groups. In our population, we observed high seropositivity in AO+ participants at all analyzed time points; however, only at 21 d did we observe a significant difference between AO+ and AO− in anti-RBD-neutralizing antibody seropositivity (Figure 5A). 

When stratified by vaccination scheme, participants who received the AZD1222, Convidecia, BNT162b2, and Sputnik V vaccines did not show significant differences in seropositivity. However, CoronaVac presented greater seropositivity in AO+ participants than in AO− participants at 21 d, both for anti-S1/S2 and anti-RBD-neutralizing IgG antibodies (Figure 5F).

### 3.7. Correlation of Anti-S1/S2 and anti-RBD-Neutralizing IgG Antibody with Participant Age

Subsequently, we evaluated whether there was a correlation between the age of participants with prior documented COVID-19. We observed a correlation between age and high levels of antibodies in previous infections and with the application of the second dose of the vaccine (90 d) in the total population. When stratifying the participants by the presence or absence of AO, we observed that only the AO− group presented a correlation between age and anti S1/S2 IgG antibodies after the second dose (90 d). Interestingly, the levels of antibodies with neutralizing activity were negatively correlated with advanced age in the longest periods (270 and 365 d). When stratifying the population by documented prior COVID-19 infection, we observed something similar to that when stratified by the presence or absence of AO, and we observed a stronger positive correlation in shorter periods (0, 21, 90, and 180 d) in those who documented previous COVID-19 (Appendix A).

### 3.8. Influence of Abdominal Obesity on the Production and Longevity of Antibodies

In the group without previous COVID-19, for those who received BNT162b2 and Sputnik V, the levels of anti-S1/S2 IgG antibodies gradually increased until reaching the first peak in sample 3 (day 90 and 2nd dose), being higher in AO+ participants, with a median of 400 AU/mL, than in the AO− participants (320 and 291 AU/mL, respectively). Interestingly, after the first dose, the AO+ AZD1222 group had a higher median of anti-S1/S2 IgG antibodies (35.7 AU/mL) than the AO− group (29 AU/mL); however, after the second dose, the median of AO− (274 AU/mL) was higher than that of AO+ (182 AU/mL) and remained so until the end of the study (126 vs. 75.8 AU/mL). 

For participants immunized with Convidecia and CoronaVac, the amount of IgG antibodies at 90 d was much lower than that in participants immunized with the other three vaccines. However, in participants who received the CoronaVac vaccine, a considerable increase was observed at 180 d (sample 4), possibly due to post-vaccination contagion, as this increase coincided with the increase in COVID-19 cases in the country. In addition, the group with the Convidecia vaccine showed an increase at 270 d (sample 5), which corresponded to the application of its reinforcement with the Moderna vaccine (Figure 6A).

In the group with previous COVID-19, the levels of IgG anti-S1/S2 antibodies in almost all vaccines showed a rapid increase after administration of the first dose (sample 2: day 21), reaching a peak of 400 AU/mL and remaining high until the end of the study. The only exception was anti-S1/S2 IgG antibodies in the AO− participants of the CoronaVac group, where the lowest levels were obtained on day 21 (mean: 172 AU/mL), followed by sample 4 (day 180), with a median of 134 AU/mL, which presented an increase in sample 5 (day 270). For the anti-RBD-neutralizing activity, we found a considerable increase in all vaccines up to the end of the study for both the AO− and AO+ groups (Figure 6B).

Finally, we again observed an increase in the levels of IgG and anti-RBD-neutralizing antibodies after 270 days, which remained high until the last sample (365 d) because the booster was administered to the entire population approximately 240 d after the first dose was administered (Figure 6).

## 4. Discussion

Abdominal obesity is a comorbidity that can influence the protective immunity generated by different vaccines against COVID-19 and could be an important factor in guiding vaccination strategies and reassessing the distribution of available vaccines in vulnerable populations [20,22]. This study reported the potential association of AO with levels of IgG anti-S1/S2 antibodies and anti-RBD-neutralizing capacity induced by documenting prior COVID-19 or via the administration of any of the five vaccination schemes applied in Mexico. To the best of our knowledge, this is the first report of AO in a Mexican population. Consequently, we showed the relationship between other variables, such as age, previous COVID-19 infection, and BMI, and the levels of antibodies generated after the start of immunization.

Obesity is a risk factor for various non-communicable diseases, such as cardiovascular diseases, diabetes, and some types of cancer [3], as well as infectious diseases, such as influenza and dengue [23,24]. In particular, AO is related to other comorbidities, such as cardiovascular disease, which is a risk factor for a poor prognosis of COVID-19 [25,26]. It is also related to respiratory distress, a high risk of hospitalization, a poor prognosis for severe COVID, and the need for ventilatory support [18,27]. This is consistent with our results, where we found a high prevalence of AO+ participants who were hospitalized and required oxygen when they were infected with SARS-CoV-2 before starting their vaccination schedules. This could be partly because the abdominal adipose tissue expresses various proinflammatory mediators that can generate a systemic inflammatory state and compromise lung function [12,13]. Furthermore, dysfunctional adipose tissue with a low adiponectin/leptin ratio can lead to increased oxidative stress and inflammation [28]. Although the lungs are the main gateway for SARS-CoV-2, there is increased expression of the angiotensin-converting enzyme type 2 (ACE2) SARS-CoV-2 receptor on host cells in the adipose tissue, which makes it a vulnerable target for COVID-19 infection [29]. Recent evidence has demonstrated the presence of the viral genome in adipose tissue samples obtained from autopsies of patients who died from severe COVID-19. The same group reported, through in vitro assays, that adipocytes and macrophages residing in adipose tissue are permissive to the virus. Additionally, these cells can generate a proinflammatory response that may participate in the development of severe COVID-19 [30]. In addition, it has been observed that ACE2 is expressed at higher levels in abdominal visceral fat than in subcutaneous fat, which would increase the production of inflammatory cytokines and their release into the systemic circulation and may contribute to an amplified “cytokine storm” in patients with abdominal obesity and high visceral adiposity [17].

Regarding the levels of antibodies induced by COVID-19, there are reports that people with obesity have higher levels of antibodies than people with normal weight, resulting in a positive correlation in the USA, Iceland, and the United Kingdom [31,32,33], which is consistent with our results, as we found a positive correlation between the levels of IgG-anti S1/S2 and anti-RBD-neutralizing antibodies with the BMI and WC of the participants. In addition, we observed elevated levels in AO+ people compared to those in AO− people, specifically in the group of participants with symptomatic disease. This disagrees with the results of a study that reported low levels in individuals with obesity in the USA [7]. Nonetheless, the underlying physiological explanation for the elevated levels of IgG and neutralizing antibodies at the post-convalescent stage remains unclear. It is important to go deeper into the study of the levels and role of antibodies against SARS-CoV-2 in people with obesity as it has been seen that obesity impairs immune function, causing chronic inflammation by increasing the number of B cells in visceral adipose tissue and producing autoreactive immunoglobulins [34] and central adiposity is associated with an increased proinflammatory fraction of IgG [35]. In addition, it has been observed that a good proportion of the antibodies produced in people with COVID-19 and obesity are autoimmune [7,36]. It has been suggested that the antibody response could be associated with secondary organ damage mediated by antibodies in addition to antiviral efficacy, which could explain why the participants in our study with higher BMI or WC had higher levels of antibodies and more severe symptoms when they had COVID-19.

In our study, vaccine-induced antibody levels were high in the AO+ group. When stratified by vaccination scheme, we observed a significant difference in anti-S1/S2 IgG antibody levels between groups that received AZD122 and CoronaVac and differences in anti-RBD-neutralizing antibodies levels between the AZD1222, Convidecia, Sputnik V, and CoronaVac groups. Interestingly, there are reports of lower antibody levels in people with high WC or abdominal obesity induced by BNT162b2 vaccines in healthcare workers in Italy [20,21] and by an inactivated vaccine in Chinese individuals [19]; both studies included people without a history of COVID-19. With this in mind, we analyzed participants without prior COVID-19 and observed elevated levels and a high seropositivity rate in the AO+ group 21 d after their first immunization. In contrast, a study on Chinese individuals observed low antibody seropositivity rates in patients with AO [19]. Interestingly, the relationship between AO and antibody levels continued independent of other variables according to our multiple linear regression analysis. However, the levels of anti-S1/S2 and anti-RBD-neutralizing IgG antibodies decreased at 180 d (sample prior to booster or third dose), especially in AO+ participants without previous COVID-19 vaccination with AZD1222 and with Covidencia, who had lower levels than the AO− group; in the case of BNT162b2 and Sputnik V this was not as evident. Sheridan et al. reported similar results, showing a correlation between BMI and elevated baseline IgG antibody levels; a higher BMI was associated with a greater decrease in influenza antibody levels in the USA 12 months after vaccination [37]. Interestingly, a study in an Israeli population observed that people with a low BMI (<18.5) vaccinated with BNT162b2 had lower antibody levels than those with a high BMI [38]. In contrast, a Greek study found no significant difference in the decrease in antibody levels at 6 months among people with different BMI vaccinated with BNT162b2 and AZD1222 [39]. It is possible that the study ethnic group could also influence the longevity of antibodies in people with obesity, as observed that among those vaccinated with BNT162b2, antibody titers were increased in Arab and Jewish ultra-orthodox individuals compared with the general population [38].

Despite the difference in vaccination-induced antibody levels between the AO+ vs. AO−, in both groups the frequency of hospitalization and use of oxygen decreased, showing that the vaccines confer protection against the severity of COVID-19 in all participants, which is similar to other studies, where the efficacy of mRNA vaccines against SARS-CoV-2 does not differ between people with obesity compared to; that in people without obesity [40,41]. However, AO+ participants decreased their antibody levels in a shorter amount of time.

When evaluating the reactogenicity induced by the five vaccines in our study, we observed that, in the AO+ and AO− groups, the administered vaccines were well tolerated, and the participants did not present serious adverse events, nor did they require hospitalization. This agrees with other studies where an inactivated COVID-19 vaccine was evaluated, finding no differences between the reactogenicity induced in people with and without obesity [19]. With the trivalent influenza vaccine, no significant differences were found in the frequency of both local and systemic reactions between people with obesity and those with normal weight [42]. The most frequent reaction was local (pain at the injection site); this is similar to other studies with inactivated vaccines [19,43] and to others where five different vaccines were evaluated (Sputnik V, AZD122, BBIBP-CorV, Convidecia, and mRNA- 1273) [44].

When analyzing the effect of obesity on BMI, a negative association between COVID-19 vaccination-induced antibody levels and BMI has been observed in Arab populations vaccinated with BNT162b2, ChAdOx-nCov-2019, and mRNA-1273, in China with an inactivated vaccine and with Coronavac in a Greek population [9,19,45]. In addition, people with severe obesity (high BMI and WC) generate significantly reduced anti-S antibody titers after vaccination with CoronaVac and BNT162b2 compared to people with normal weight [8]. Interestingly, BMI has no significant association with antibody levels induced by the BNT162b2 vaccine in a New Zealand population [46] or in US health workers [47]. Moreover, a study from Japan observed that anti-SARS-CoV-2 IgG titers tended to decrease with increasing BMI in men and did not differ significantly between BMI categories in women vaccinated with BNT162b2 [48]. Our results showed a positive correlation between BMI and vaccination-induced antibody levels at 21, 90, 270, and 365 d for IgG anti-S1/S2 IgG and at 21 d for anti-RBD-neutralizing antibodies. This suggests that various factors must be considered, such as sex and study population. In contrast, when compared with WC, we observed a positive correlation at 21, 90, 270, and 365 d for IgG anti-S1/S2 levels and at 21 and 90 d for anti-RBD-neutralizing antibodies; this could be because AO is a more sensitive parameter than BMI for predicting the prognosis of COVID-19 [17,18]. Therefore, we propose that the use of both WC and BMI could complement the evaluation of the effects of obesity on antibody levels.

Documentation before COVID-19 is associated with high levels of antibodies against SARS-CoV-2 induced by vaccination [49,50]. Our results showed that participants with prior COVID-19 exhibited significantly high antibody levels from the first vaccination dose and remained so until 180 d of the study, except for AO+ participants who received the CoronaVac vaccine. Similarly, Gobbi et al. reported higher titers of IgG and neutralizing antibodies in participants with prior COVID-19 from the first dose of the vaccine than in those without prior COVID-19, who reached the same level with the second dose [51] and presented the broadest neutralizing activity against SARS-CoV and variants of SARS-CoV-2 [52]. In addition, it has been observed that symptomatic people have higher antibody titers than asymptomatic people without previous COVID-19 [53,54].

Our results did not show an association between the levels of antibodies induced by the COVID-19 vaccination in patients with hypertension; however, we observed a negative independent association with T2D only at 270 d. However, this effect may be due to the small number of participants with these characteristics, as hypertension has been reported to negatively influence vaccine-induced antibody levels, and convalescent individuals with metabolic syndrome comorbidities have significantly high number of antibodies [20,31,50]. Therefore, it would be of interest to evaluate a larger number of patients with these comorbidities.

We observed an association between the type of vaccination scheme and antibody levels, independent of the other variables analyzed in this study. In addition, AO− and AO+ participants without previous COVID-19, vaccinated with BNT162b2 and Sputnik V, reached the highest antibody levels after the second dose, and the lowest levels in those immunized with CoronaVac. When evaluating AO− and AO+ individuals with prior COVID-19, we found that all vaccines induced very high levels of antibodies after the first vaccine dose, except in patients with AO− immunized with CoronaVac. Similarly, participants with severe obesity (with and without prior COVID-19) and normal weight (with and without prior COVID-19) vaccinated with BNT162b2 showed significantly higher levels than those vaccinated with CoronaVac [8]. Another study showed that the BNT162b2 vaccine induced a robust immune response against SARS-CoV-2 and its variants, whereas CoronaVac was the least effective, leaving Sputnik V, AZD122, and Convidecia with intermediate responses [52]. Finally, participants vaccinated with BNT162b2, with and without prior COVID-19, had a higher neutralizing capacity than those immunized with CoronaVac and Convidecia [49].

Finally, we analyzed the association of age and observed a positive correlation with the levels of antibodies induced by COVID-19 and by vaccination in the AO− group. However, the AO+ group showed an association only at 90 d after vaccination. Similar to our results, a study of participants with obesity found no correlation between antibody levels and age after the second dose of the mRNA vaccine [20]. Additionally, the correlation was stronger in participants with documented COVID-19 than in those without. Interestingly, there are reports of participants without prior COVID-19, where they observed a negative correlation between age and anti-S1 IgG antibody levels induced by the BNT162b2 vaccine and CoronaVac [8] and significantly low anti-S IgG titers in older participants immunized with the BNT162b2 vaccine [55]. In addition, a stronger antibody response was observed in vaccinated individuals aged 25–50 years than in those aged 80–95 [56]. The association in our study remained regardless of whether a multiple regression analysis was performed. However, when stratified by the vaccination scheme, we did not find a correlation, which is consistent with another study in Mexico [49]. This could be due to the fact that vaccination in Mexico was conducted by age group, preventing comparison in a broad range of years and that our cohort had a small number of older people.

The strength of our study lies in the evaluation of the effect of AO and BMI on the levels of antibodies induced by five different vaccines, their reactogenicity, and their protective effect in the Mexican AO+ population. Additionally, our study highlighted the necessity to evaluate its effect on the levels of antibodies and COVID-19 symptoms as it has been observed that AO is a more sensitive parameter for the prediction a negative result with COVID-19 than BMI and subcutaneous fat [17,18]. However, based on our results, we suggest the use of BMI and WC to fully assess the effects of obesity on infectious diseases and vaccination, especially in Mexico, where the prevalence of obesity is very high. One limitation of our study is the small number of participants with some comorbidities, which did not allow us to perform a more detailed analysis of their influence on antibody response, in addition to having a small number of participants aged > 60 years.

This study raises questions about the inflammatory state of the participants, which can better explain the relationship between AO and the levels of antibodies induced by COVID-19 and vaccination. It would also be interesting to investigate the cellular response of the AO+ participants and whether this response is impaired. Finally, the effects of a heterologous vaccination schedule over a long period in the AO+ participants were determined.

## 5. Conclusions

In conclusion, we found that AO is related to a worse prognosis for COVID-19, with higher levels of antibodies induced by COVID-19 and vaccination, especially in cases where prior COVID-19 was documented. AO did not influence the reactogenicity of the five evaluated vaccines, which provided protection against SARS-CoV-2 infection, reducing severe symptoms of COVID-19 and resulting in safe vaccination schemes for the population studied. In addition, factors such as age, type of vaccine, BMI, and prior COVID-19 are important factors in the humoral response to COVID-19.

## Figures and Tables

**Figure 1 vaccines-12-00088-f001:**
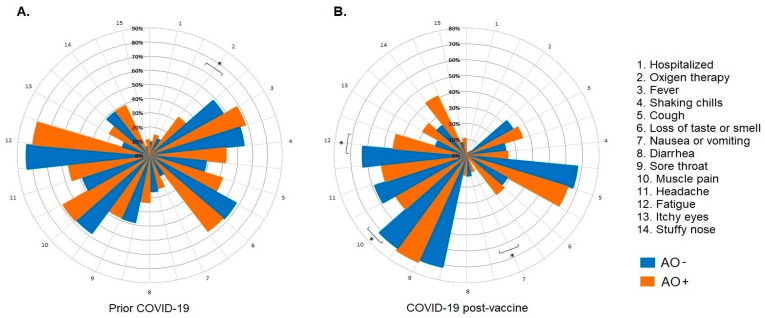
Clinical profiles by (**A**) prior COVID-19, (**B**) COVID-19 post-vaccination in all populations. Each graph shows the proportion, from 0% (center of the circular graph) to 100% (circular graph perimeter). The * *p*-value refers to the chi-square/Fisher’s exact test. The results were considered statistically significant when *p* < 0.05. Abbreviations: AO−, without abdominal obesity; AO+, with abdominal obesity.

**Figure 2 vaccines-12-00088-f002:**
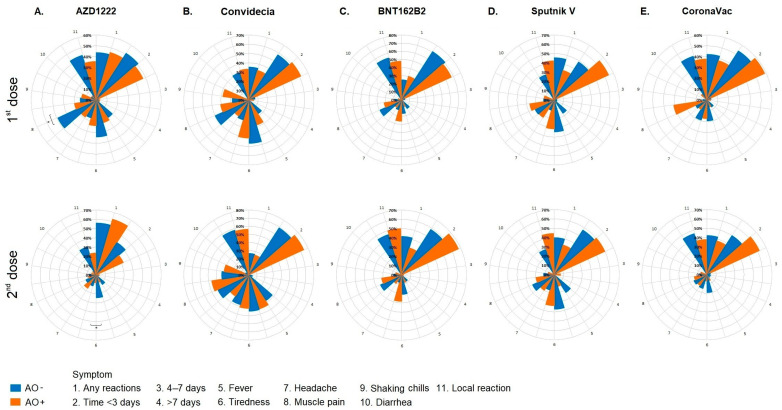
Clinical profiles by vaccination schedule in all population; (**A**) AZD1222, (**B**) Convidecia, (**C**) BNT162B2, (**D**) Sputnik V, and (**E**) CoronaVac. Each graph shows the proportion, from 0% (center of the circular graph) to 100% (circular graph perimeter). The * *p*-value refers to the chi-square/Fisher’s exact test. The results were considered statistically significant when *p* < 0.05. Abbreviations: AO−, without abdominal obesity; AO+, with abdominal obesity.

**Figure 3 vaccines-12-00088-f003:**
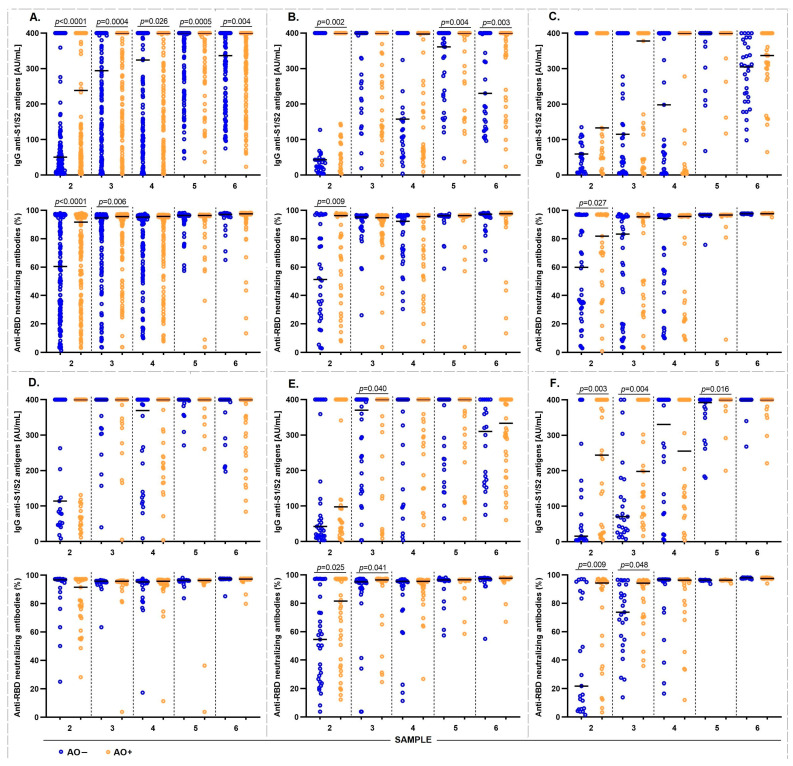
Anti-S1/S2 and anti-RBD neutralizing IgG antibody levels in AO− and AO+ participants with abdominal obesity induced by the vaccination scheme received. Data from the three follow-up time points: sample 2 (day 21), 3 (day 90), 4 (day 180), 5 (day 270), and 6 (day 365). Comparison of levels of IgG anti-S1/S2 antibodies and neutralizing anti-RBD neutralizing antibodies between AO− and AO+ participants by vaccination scheme received: (**A**) total population, (**B**) AZD1222, (**C**) Convidecia, (**D**) BNT162b2, (**E**) Sputnik V, and (**F**) CoronaVac. Points, individuals; bars medium; comparisons using the Mann–Whitney U test. The results were considered statistically significant when *p* < 0.05. The colors group the participants into AO− (blue) and AO+ (orange).

**Figure 4 vaccines-12-00088-f004:**
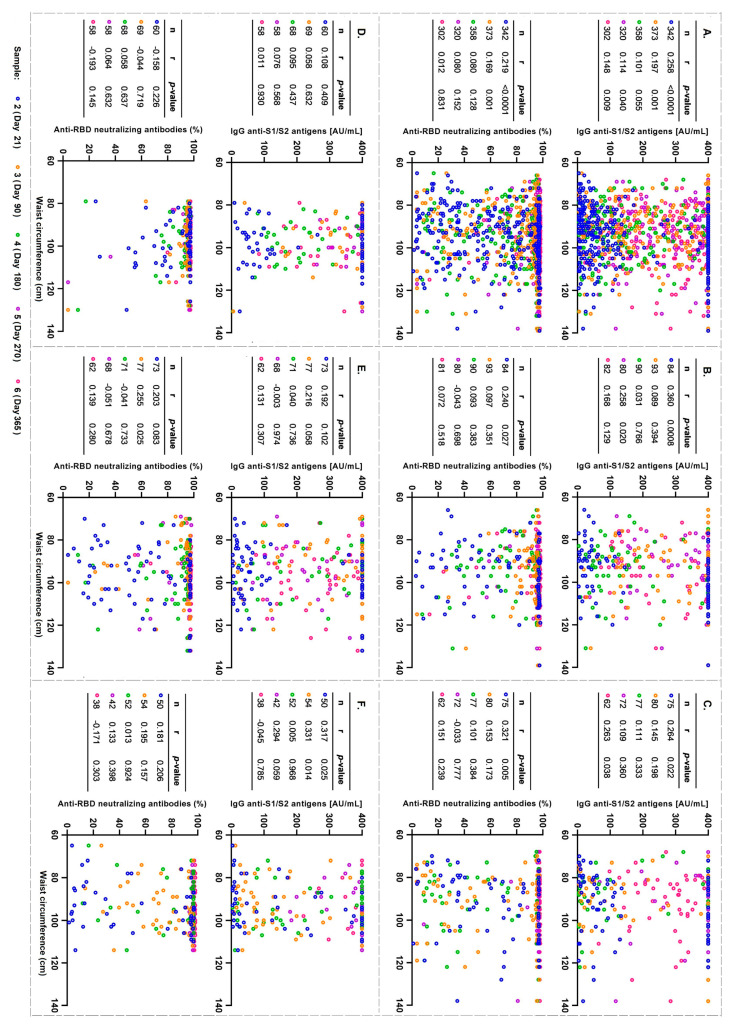
Correlation of waist circumference with the levels of anti-S1/S2 IgG antibodies and anti-RBD neutralizing antibodies induced by the vaccination scheme. Data from the three follow-up time points: sample 2, 3, 4, 5, and 6. Correlation by vaccination scheme: (**A**) total population, (**B**) AZD1222, (**C**) Convidecia, (**D**) BNT162b2, (**E**) Sputnik V, and (**F**) CoronaVac. r denotes Spearman’s correlation coefficient.

**Figure 5 vaccines-12-00088-f005:**
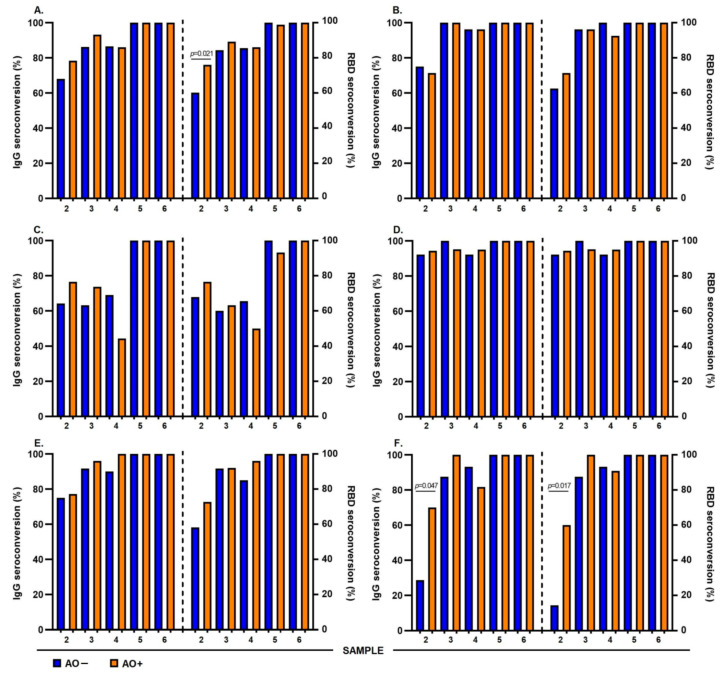
Seroconversion rates of IgG anti-S1/S2 and neutralizing anti-RBD neutralizing antibodies in AO− (blue) and AO+ (orange). Data are taken from follow-up time points: samples 2, 3, 4, 5, and 6 (21, 90, 180, 270, and 365 d, respectively). (**A**) Total population; participants AO+ have significantly higher anti-RBD neutralizing antibody seropositivity than AO− at 21 d. Vaccine (**B**) AZD1222, (**C**) Convidecia, (**D**) BNT162b2, (**E**) Sputnik V, and (**F**) CoronaVac, participants AO+ have significantly higher IgG anti-S1/S2 and anti-RBD neutralizing antibody seropositivity than AO− at 21 d. Bars medium; comparisons using the Mann–Whitney U test. The results were considered statistically significant when *p* < 0.05.

**Figure 6 vaccines-12-00088-f006:**
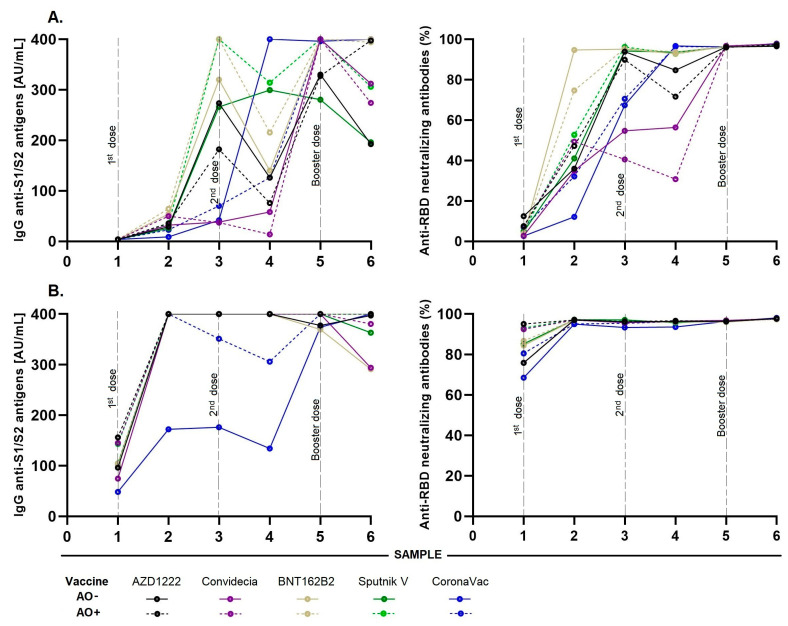
Analysis of the longevity of the levels of anti-S1/S2 IgG antibodies and anti-RBD neutralizing antibodies with neutralizing capacity induced by the different vaccines: (**A**) Median antibody levels in participants without history of COVID-19; participants in the AZD1222, Sputnik V and CoronaVac group without previous COVID-19 had significant differences in their antibody levels between AO+ and AO− (21, 360; 21, 90; 21, 90 d, respectively). (**B**) With a history of prior COVID-19; participants in the AZD1222 and Convidecia group with previous COVID-19 had significant differences in their antibody levels between AO+ and AO− (0, 180; 360 d, respectively). Comparisons using the Mann–Whitney U or Kruskal–Wallis tests, as appropriate. Abbreviations: AO−, without abdominal obesity; AO+, with abdominal obesity.

**Table 1 vaccines-12-00088-t001:** Clinical demographic characteristics of the study participants.

	AZD1222		Convidecia		BNT162B2		Sputnik V		CoronaVac	
	AO−	AO+	AO−	AO+	AO−	AO+	AO−	AO+	AO−	AO+
	(n = 41)	(n = 59)	(n = 45)	(n = 39)	(n = 24)	(n = 46)	(n = 35)	(n = 45)	(n = 26)	(n = 29)
Female	24 (58.5)	41 (69.5)	35 (77.8)	28 (71.8)	10 (41.7)	36 (78.3)	14 (40)	34 (75.6)	16 (61.5)	19 (65.5)
Male	17 (41.5)	18 (30.5)	10 (22.2)	11 (28.2)	14 (58.3)	10 (21.7) c **	21 (60)	11 (24.4) c **	10 (38.5)	10 (34.5)
Age years	50 (36–54)	53 (41–56) *	40 (35–44)	43 (37–48)	54 (51–57.5)	54.5 (52–57)	42 (29–56)	53 (50–56) *	35.5 (33–37)	37 (33–38)
Weight (kg)	62.9 (55–73)	76 (67–90) ***	63 (57–70)	80 (70–97) ***	67 (58–74)	76.5 (69–85) ***	67 (61–75)	78 (74–84) ***	64 (57–82)	79 (73–85) **
Height (m)	1.6 (1.52–1.69)	1.6 (1.52–1.66)	1.6 (1.52–1.65)	1.6 (1.57–1.71)	1.6 (1.55–1.70)	1.6 (1.53–1.65)	1.7 (1.6–1.7)	1.6 (1.54–1.68) **	1.6 (1.56–1.68)	1.6 (1.58–1.68)
WC (m)	0.85 (0.78–0.90)	1.02 (0.93–1.10) ***	0.82 (0.79–0.86)	1.03 (0.95–1.11) ***	0.91 (0.84–0.96)	1.03 (0.96–1.09) ***	0.87 (0.81–0.92)	1.03 (0.96–1.08) ***	0.86 (0.78–0.93)	1 (0.95–1.04) ***
HC (m)	0.98 (0.95–1.02)	1.1 (1.03–1.18) ***	0.97 (0.92–1.03)	1.11 (1.06–1.24) ***	0.97 (0.95–1.00)	1.11 (1.06–1.15) ***	0.96 (0.92–1.03)	1.14 (1.08–1.19) ***	1.03 (1.00–1.06)	1.11 (1.06–1.16) ***
Waist/hip ratio	0.87 (0.81–0.91)	0.93 (0.87–0.96) ***	0.84 (0.80–0.89)	0.9 (0.87–0.96) ***	0.94 (0.87–0.98)	0.92 (0.88–0.97)	0.9 (0.84–0.96)	0.92 (0.87–0.94)	0.84 (0.78–0.89)	0.9 (0.84–0.94) **
BMI	25.3 (23.6–26.4)	29.4 (27.6–32.9) ***	25.2 (22.8–28)	29.7 (26.5–35.5) ***	25.3 (23.3–27.6)	30.1 (27.9–32) ***	24.2 (23.1–27.1)	30.1 (27.7–32.9) ***	27 (22.9–29.1)	30.3 (26.9–32.5) **
Normal weight	18 (43.9)	2 (3.4)	22 (48.9	5(12.8)	11 (45.8)	2 (4.3)	23 (65.7)	3 (6.7)	10 (38.5)	3 (10.3)
Overweight	21 (51.2)	30 (50.8)	18 (40)	16 (41)	13 (54.2)	19 (41.3)	11 (31.4)	19 (42.2)	10 (38.5)	10 (34.5)
Obesity	2 (4.9)	27 (45.8) a ***	5 (11.1)	18 (46.2) a ***	0 (0)	25 (54.3) a ***	1 (2.9)	23 (51.1) a ***	6 (23.1)	16 (55.2) a *
T2D	4 (9.8)	12 (20.3)	1 (2.2)	2 (5.1)	6 (25)	6 (13)	3 (8.6)	9 (20)	1 (3.8)	0 (0)
HTA	4 (9.8)	19 (32.2) *	0 (0)	3 (7.7)	4 (16.7)	13 (28.9)	4 (11.4)	12 (26.7)	0 (0)	2 (6.9)
AD	5 (12.2)	1 (1.7) *	2 (4.4)	2 (5.1)	0 (0)	2 (4.3)	2 (5.7)	1 (2.2)	0 (0)	0 (0)
ORD	4 (9.8)	6 (10.2)	2 (4.4)	1 (2.6)	2 (8.3)	4 (8.7)	0 (0)	0 (0)	0 (0)	1 (3.4)
HD	0(0)	4 (6.8)	0 (0)	2 (5.1)	3 (12.5)	1 (2.2)	0 (0)	0 (0)	0 (0)	0 (0)
HIV	1 (2.4)	0 (0)	0 (0)	0(0)	0 (0)	0 (0)	1 (2.9)	0 (0)	0 (0)	0 (0)
Seronegative	29 (70.7)	28 (47.5)	31 (68.9)	19 (48.7)	13 (54.2)	21 (45.7)	25 (71.4)	26 (57.8)	17(65.4)	11 (37.9)
Asymptomatic	3 (7.3)	15 (25.4)	6 (13.3)	8 (20.5)	7 (29.2)	18 (39.1)	4 (11.4)	9 (20)	5 (19.2)	9 (31)
Symptomatic	9 (22)	16 (27.1) b *	8 (17.8)	12 (30.8)	4 (16.7)	7 (15.2)	6 (17.1)	10 (22.2)	4 (15.4)	9 (31)
Smoking history										
Never	29 (70.7)	50 (84.7)	41 (91.1)	33 (84.6)	21 (87.5)	38 (82.6)	31 (88.6)	39 (86.7)	22 (84.6)	23 (79.3)
Mild	4 (9.8)	5 (8.5)	2 (4.4)	3 (7.7)	1 (4.2)	5 (10.9)	0 (0)	3 (6.7)	2 (7.7)	4 (13.8)
Moderate	4 (9.8)	2 (3.4)	1 (2.2)	3 (7.7)	1 (4.2)	1 (2.2)	3 (8.6)	1 (2.2)	1 (3.8)	1 (3.4)
Severe	4 (9.8)	2 (3.4)	1 (2.2)	0 (0)	1 (4.2)	2 (4.3)	1 (2.9)	2 (4.4)	1 (3.8)	1 (3.4)
Alcoholism										
Never	25 (61)	43 (72.9)	26 (57.8)	31 (79.5)	19 (79.2)	32 (69.6)	22 (62.9)	37 (82.2)	16 (61.5)	19 (65.5)
Mild	13 (31.7)	15 (25.4)	17 (37.8)	7 (17.9)	5 (20.8)	13 (28.3)	11 (31.4)	8 (17.8)	10 (38.5)	9 (31)
Moderate	3 (7.3)	1 (1.7)	2 (4.4)	1 (2.6)	0 (0)	1 (2.2)	1 (2.9)	0 (0)	0 (0)	0 (0)
Severe	0 (0)	0 (0)	0 (0)	0 (0)	0 (0)	0 (0)	1 (2.9)	0 (0)	0 (0)	1 (3.4)

Data are presented as numbers (%) or median (interquartile range). The *p* value refers to the chi-square/Kruskal–Wallis test, the letter indicates that there is a comparison with a statistically significant difference of participants AO− vs. AO+: (a) Between the groups classified by BMI (normal weight, overweight, or obese). (b) Between the groups (seronegative, asymptomatic, or symptomatic). (c) By sex (male or female). Abbreviations: kg, kilogram; m, meter; WC, waist circumference; HC, hip circumference; BMI, body mass index; HIV, human immunodeficiency virus; HTA, hypertension; T2D, type 2 diabetes; AD, autoimmune disease; ORD, other respiratory disease; HD, heart disease. Seroprevalence was categorized as prior COVID-19 (symptomatic and asymptomatic), and seronegative. Body mass index was categorized as normal weight (<24.9 kg/m^2^), overweight (25 to 29.9 kg/m^2^), and obesity (>30 kg/m^2^). Abdominal obesity was categorized by a waist circumference greater than 1.02 m in men and 0.86 m in women. * *p* < 0.05, ** *p* ≤ 0.01), *** *p* ≤ 0.001.

**Table 2 vaccines-12-00088-t002:** Influence of abdominal obesity on the antibody response induced by vaccines in individuals without and with COVID-19 prior to vaccination.

			Anti-S1/S2 IgG Antibodies Production (AU/mL)			Anti-RBD Neutralizing Antibodies (%)		
	Day	N	AO−	AO+	*p*-Value	AO−	AO+	*p*-Value
All (seropositive)	0	168	83.8	137.5	0.001	76.9	91.3	0.002
Asymptomatic	0	84	56.7	95.1	0.139	72.1	78.9	0.146
Symptomatic	0	84	90.7	188.5	<0.0001	81.8	94.5	0.001
Without prior COVID-19	0	220	3.8	3.8	0.158	4.95	5.58	0.874
	21	190	31.7	45.4	0.011	35.95	55.27	0.013
	90	211	187	243	0.127	90.22	93.6	0.167
	180	204	135	177	0.854	92	91	0.706
	270	185	400	400	0.365	96.38	96.37	0.349
	365	164	338	351	0.932	97.50	97.54	0.777
With prior COVID-19	0	168	90.65	138	0.001	79.13	91.27	0.002
	21	151	400	400	0.114	96.94	96.96	0.924
	90	161	400	400	0.583	95.99	96.06	0.795
	180	153	400	400	0.236	96	96	0.172
	270	134	400	400	<0.0001	96.5	96.41	0.199
	365	137	331.5	400	0.002	97.65	97.65	0.099

The Mann–Whitney U test or Kruskal–Wallis test was used to evaluate differences across the groups. The results were considered statistically significant when *p* < 0.05. Abbreviations: AO−, without abdominal obesity; AO+, with abdominal obesity.

**Table 3 vaccines-12-00088-t003:** Analysis of the association of anthropometric factors with the anti-S1/S2 and anti-RBD IgG antibody neutralizing response in the study population.

**IgG Anti-S1/S2 AU/mL**												
	**Day 0**		**Day 21**		**Day 90**		**Day 180**		**Day 270**		**Day 365**	
	***p*-Value**	**Beta**	***p*-Value**	**Beta**	***p*-Value**	**Beta**	***p*-Value**	**Beta**	***p*-Value**	**Beta**	***p*-Value**	**Beta**
Prior COVID-19	<0.0001	0.261	<0.0001	0.826	<0.0001	0.483	<0.0001	0.481	<0.0001	0.289	<0.0001	0.281
Abdominal obesity	0.005	0.208	0.001	0.099	0.461	0.034	0.46	0.035	0.161	0.077	0.066	0.104
Age	0.018	0.176	0.244	0.035	<0.0001	0.223	0.035	0.099	0.483	0.038	0.074	−0.1
Sex	0.258	−0.084	0.825	−0.006	0.897	0.006	0.87	−0.008	0.006	−0.147	0.8	−0.014
T2D	0.23	0.09	0.994	0	0.969	0.002	0.892	0.006	0.986	0.001	0.78	0.015
Hypertension	0.626	0.038	0.91	−0.003	0.931	0.004	0.654	−0.022	0.112	−0.085	0.923	0.005
Smoking history	0.745	0.024	0.5	0.02	0.31	0.045	0.704	0.018	0.323	0.053	0.31	−0.057
Alcoholism	0.409	−0.061	0.743	0.01	0.649	0.02	0.245	0.054	0.633	−0.026	0.025	0.124
BMI class	0.334	0.081	0.393	0.03	0.676	0.019	0.125	−0.072	0.951	0.003	0.415	0.046
Vaccine	0.247	−0.085	0.08	−0.051	0.091	−0.075	0.014	0.115	0.113	0.085	0.002	0.171
**Anti-RBD neutralizing antibodies (%)**												
	**Day 0**		**Day 21**		**Day 90**		**Day 180**		**Day 270**		**Day 365**	
	***p*-value**	**Beta**	***p*-value**	**Beta**	***p*-value**	**Beta**	***p*-value**	**Beta**	***p*-value**	**Beta**	***p*-value**	**Beta**
Prior COVID-19	0.027	0.167	<0.0001	0.61	<0.0001	0.321	<0.001	0.342	0.992	0.001	0.275	0.063
Abdominal obesity	0.314	0.088	0.001	0.137	0.336	0.049	0.328	0.049	0.903	0.007	0.617	0.029
Age	0.006	0.208	0.078	0.075	0.001	0.097	0.196	0.065	0.03	−0.122	0.001	−0.186
Sex	0.322	−0.074	0.409	−0.035	0.801	0.012	0.815	−0.012	0.43	−0.044	0.64	0.027
T2D	0.342	0.073	0.565	0.024	0.656	−0.022	0.585	−0.027	0.005	−0.158	0.862	−0.01
Hypertension	0.464	0.059	0.669	−0.018	0.566	0.03	0.784	−0.014	0.751	−0.019	0.568	−0.034
Smoking history	0.899	0.01	0.704	−0.016	0.842	−0.01	0.53	0.031	0.677	0.023	0.989	0.001
Alcoholism	0.765	0.022	0.468	−0.031	0.627	−0.024	0.477	0.036	0.4	−0.047	0.293	−0.061
BMI class	0.026	0.169	0.489	0.035	0.265	0.056	0.665	−0.022	0.967	−0.002	0.183	0.077
Vaccine	0.482	−0.053	0.164	−0.058	0.798	0.013	0.015	0.122	0.954	0.003	0.064	0.107

For each regression analysis, the results are presented in terms of the beta values. The *p*-value refers to a multivariate linear regression analysis. The results were considered statistically significant when *p* < 0.05. Previous COVID-19 was categorized as follows: without prior COVID-19 or with prior COVID-19; asymptomatic or symptomatic. Abdominal obesity was categorized as follows: AO− and AO+. BMI class was categorized as follows: normal weight (<24.9 kg/m^2^), overweight (25–29.9 kg/m^2^), or obese (>30 kg/m^2^). Vaccine was categorized as follows: AZD1222, Convidecia, BNT162b2, Sputnik V, or CoronaVac. Abbreviations: T2D, Type 2 diabetes; BMI, body mass index.

## Data Availability

The data set(s) supporting the results of this article are included within this article.

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
