# Peer review of "A Potential Association between Abdominal Obesity and the Efficacy of Humoral Immunity Induced by COVID-19 and by the AZD1222, Convidecia, BNT162b2, Sputnik V, and CoronaVac Vaccines"

_vaccines, 2024, doi:10.3390/vaccines12010088_

Round 1

Reviewer 1 Report

Comments and Suggestions for Authors

In this study, researchers aim to understand the association of abdominal obesity with the efficacy of antibody responses to COVID-19 vaccines. However, the manuscript is poorly written, it is unclear, very difficult to follow due to poor organization and there are misleading statements in results and conclusions.

1.       The manuscript lacks clarity and is poorly organized. In the abstract authors mentioned that they wanted to measure antibody responses for 1 year after vaccination in individuals with prior COVID-19 or COVID-19 post-vaccination. They also wanted to test the association of abdominal obesity with antibody responses and compare the durability of antibody responses in individuals with or without abdominal obesity. However, the way the manuscript is written, it is difficult to follow the data and understand what they observed.

2.       The study cohort is not clear. Was it a prospective study cohort? What were the inclusion and exclusion criteria? Was it a home visit or a hospital follow-up?

3.       Regarding ‘reactogenicity’, this word is misleading. Do they mean ‘self-reported adverse events’ to the COVID-19 vaccine or something else? In figure 1, the authors show the self-reported adverse events in people who had COVID-19 (A) prior or (B) post-vaccination. It is not clear when these responses were recorded. Likewise, it is not clear whether figures C to G include participants having prior or post-COVID-19 or not.

4.       Most importantly, how were the self-reported adverse events recorded? Adverse events are one of the major components of this paper and yet the authors have not mentioned when and how these data were collected.

5.       The definition for those with prior COVID-19 is also misleading or unclear. Did the researchers use hospital records, e.g., RT-PCR positive for SARS-CoV-2, to define confirmed COVID-19 cases or they just relied on participants response? If not confirmed with hospital records, these results can be misleading, specially, authors mentioned they defined prior COVID-19 even based on ‘clinical symptomatology’. There is no way to define COVID-19 versus other respiratory diseases just based on ‘clinical symptomatology’.

6.       What is being measured in the ‘neutralizing antibody’ measurement the authors are referring to?  There are discrepancies throughout the manuscript on the way this result is presented, like ‘neutralizing antibody’, anti-RBD IgG, or even RBD. Neutralizing antibody, in the correct sense, refers to result from live virus neutralization assay. There are pseudo-virus neutralizing antibody assays as well as assays like ACE2 inhibition antibody assays. Authors need to be clear on what they measured and write it correctly as the results of these different assays will be different and carry different meanings. Moreover, was the S1/S2 antibody response as well as this author-referred ‘neutralizing antibody’ response against ancestral strain or other variants? The changes should be made consistently on figure and table legends as well.  

7.       The authors need to cite the methods they followed and used to define abdominal obesity based on waist circumference.

8.       While presenting results, authors need to provide a flow and present data in sequential order. E.g., after reactogenicity, they directly jumped into multiple linear regression. I suggest they discuss the antibody data first, on which they used multiple linear regression analysis.

9.       While explaining the results, it is better to explain briefly what they did and what they are showing.

10.   In the result section 3.2, it is not clear how they decided on positive versus negative association. In this result section, authors mentioned ‘an additional model’, where the data is and what are they talking about is not clear.

11.   In the result section, in several instances, the authors mention there was an association, but it is not clear what kind of association it was.

12.   The authors go back and forth to prior COVID-19 and vaccination results that makes again very difficult to follow what they are showing. My suggestion is to explain the findings from the cohort with ‘prior COVID-19’ first (e.g., figure 1, table 2, table 3) and then only show all the results from ‘vaccination’.

13.   Figure and table legends should stand alone and be understandable. Add more details about the experimental groups, procedures performed, data shown (e.g., mean, median etc.), and statistical tests used in each figure and table legends. Figures and tables need to be explained and cited in the text which is not done. E.g., figures 1D_G are not discussed anywhere.

14.   In result section 3.3., and elsewhere authors mention ‘behavior’ to refer to differences in antibody response at different time points. Researchers are not reporting ‘behavior’ here. Use appropriate words.

15.   In section 3.6, authors mentioned ‘correlation of IgG and RBD with participant age’. RBD is an antigen while IgG is an antibody. Researchers did not measure RBD antigen quantity rather they might have measured some sort of antibody response they are referring to as ‘neutralizing antibody’. This is completely wrong and misleading.

16.   Authors need to define abbreviations at their first use. There are several ones not defined before their first use, including IMSS, WAO, and WoAO.

17.   Extensive English language editing is recommended. There are typos everywhere, including figure legends, figure keys, and text.

18.   How does this paper differ from author's prior publication: https://www.ncbi.nlm.nih.gov/pmc/articles/PMC10453006/

19.   The argument authors put forward that patients with abdominal obesity had less durable antibody response does not look correct as the antibody response up to 6 time points (figure 5) between patients with WoAO and WAO looks similar. There is no statistics used or shown indicating clear differences in the durability of antibodies.

20.   In discussion, lines 387-390, the author's conclusion that autoimmune antibodies explain the higher antibody response in patients with higher BMI or WC in their study is mere a hypothesis and they do not have any concrete evidence to support this. They need to be careful in its interpretation.

21.   The results are in contrary to several previous studies, except for the one that the same group published earlier most probably using overlapping data (?). However, the authors fail to extrapolate what might be different between studies so that the results are different.

22.   The conclusion that AO is associated with a worse prognosis of COVID-19, is again irrelevant to this study. This is an antibody response study, not an infection study where there is a prognosis of COVID19 investigated. Likewise, they mentioned COVID-19 vaccine reduces symptoms of COVID-19 in individuals with AO, which is again not what their research shows. There are not COVID-19 symptom studied in here, they just looked into adverse response to vaccines. 

Comments on the Quality of English Language

Extensive English language editing is recommended. There are typos everywhere, including figure legends, figure keys, and text.

Author Response

1. The results section was restructured as you suggested, and further explanations were added in points 8 and 12. A total of 6 figures and 8 sections of results were restructured.

2. The answer was added in section 2.2 as follows:

A prospective, longitudinal cohort study was conducted to evaluate the immuno-logical status of the Mexican population before and after the application of any of the five vaccination schemes against SARS-CoV-2. Participants were recruited between May 14 and July 27, 2021, and were monitored during hospital follow-up for 12 months after the first dose of the vaccine was administered. A total of three hundred eighty-nine participants were included, and the number of participants recruited for each vaccination scheme was as follows: 100 from AZD1222, 84 from Convidecia, 70 from BNT162b2, 80 from Sputnik V, and 55 from CoronaVac. The inclusion criteria were as follows: adult participants without previous application of any vaccination scheme against COVID-19 and who agreed to sign an informed consent form. The exclusion criteria were presence of symptoms or signs of respiratory disease and refusal to sign an informed consent form.

3. The answer was added in section 2.2 as follows:

Self-reported adverse events or reactogenicity (physical manifestations of the in-flammatory response to vaccination: fever, tiredness, headache, muscle pain, shaking chills, diarrhea, and local reaction) were collected using a questionnaire on day 21 after the first dose (sample 2) and on day 90 (sample 3) after the application of the second dose of each vaccination schedule, except for the vaccination schedule, Convidecia, which was at 270 days (Moderna vaccine booster).

The figure 1C-1G includes the all-study population, those who had prior COVID-19 and those who had not prior COVID-19 and is added in the Figure 1 legend.

4. The answer was added in section 2.2 as follows:

were collected using a questionnaire on day 21 after the first dose (sample 2) and on day 90 (sample 3) after the application of the second dose of each vaccination schedule, except for the vaccination schedule, Convidecia, which was at 270 days (Moderna vaccine booster).

5.  At each participant visit for sample collection, they were asked for proof of vaccination and a positive PCR test if they had prior COVID-19 to their first dose of vaccine to fill out the data in the questionnaires. We use the clinical symptoms to complement the results of the PCR or antibody tests that were performed at the beginning of the study.

6. As mentioned in your comment, neutralization tests are based on the use of live viruses and the use of cell culture or, failing that, pseudoviruses, however this requires a biosafety level (BSL) 3 laboratory for wild type SARS-CoV-2, therefore, the use of Elisa assays is highly recommended. The neutralizing activity of the Anti-RBD neutralizing antibodies was assessed using an ELISA SARS-CoV-2 surrogate virus neutralization test kit (GenScript, NJ, USA; catalog number L00847). The kit contains the cell surface receptor ACE2 immobilized on 96-well plates and includes an analog to the horseradish peroxidase (HRP)-labeled receptor binding domain (HRP-RBD), which was previously incubated with the serum to be tested. The neutralizing antibodies in the serum bind to HRP-RBD and block its interaction with the cellular receptor ACE2. The tests were performed following the manufacturer’s recommendations, and negative and positive controls were used to validate the results. The cutoff values were as follows: negative < 30% inhibition and positive ≥ 30% inhibition. This assay has a high specificity of 99.2 (95%CI: 96.9–99.9) and overall sensitivity of 80.3 (95%CI: 74.9–84.8) for the surrogate virus neutralization test (sVNT). The antibodies are against the antigen of a Wild-Type SARS-CoV-2 and Variants of Concern (VOC) including Alpha, Beta, Gamma, Delta.

And the correct name was unified throughout the text, figures and tables (Anti-RBD neutralizing antibodies).

Reference:

  • Alexandra Rockstroh, Johannes Wolf, Jasmin Fertey, Sven Kalbitz, Stefanie Schroth, Christoph Lübbert, Sebastian Ulbert & Stephan Borte. (2021) Correlation of humoral immune responses to different SARS-CoV-2 antigens with virus neutralizing antibodies and symptomatic severity in a German COVID-19 cohort. Emerging Microbes & Infections 10:1, pages 774-781
  • Jung J, Rajapakshe D, Julien C, Devaraj S. Analytical and clinical performance of cPass neutralizing antibodies assay. Clin Biochem. 2021 Dec; 98:70-73. doi: 10.1016/j.clinbiochem.2021.09.008. Epub 2021 Sep 21. PMID: 34560062; PMCID: PMC8453782.
  • Papenburg J, Cheng MP, Corsini R, Caya C, Mendoza E, Manguiat K, Lindsay LR, Wood H, Drebot MA, Dibernardo A, Zaharatos G, Bazin R, Gasser R, Benlarbi M, Gendron-Lepage G, Beaudoin-Bussières G, Prévost J, Finzi A, Ndao M, Yansouni CP. Evaluation of a Commercial Culture-Free Neutralization Antibody Detection Kit for Severe Acute Respiratory Syndrome-Related Coronavirus-2 and Comparison With an Antireceptor-Binding Domain Enzyme-Linked Immunosorbent Assay. Open Forum Infect Dis. 2021 Apr 30;8(6):ofab220. doi: 10.1093/ofid/ofab220. PMID: 34136587; PMCID: PMC8135688.

7. The answer was added in section 2.4 as follows: The participants' WC was measured in meters (m) at the midpoint between the last rib and the iliac crest in the patient standing and exhaling, participants with a WC > 1.02 m in men and > 0.88 m in women, they were defined as having abdominal obesity (AO+) and participants with a circumference below these values were defined as participants without abdominal obesity (AO-), according to the thresholds proposed by the National Institute of Health (NIH).

8. We present the data in sequential order, first the "3.2. Anti-S1/S2 and anti-RBD-neutralizing IgG antibody response induced by COVID-19 in AO+ individuals" point, which includes antibodies and symptoms induced by COVID-19. Subsequently, "3.3. Reactogenicity of the different vaccination schedules applied in the population with AO". Afterwards, the multiple linear regression, “3.4. Prior COVID-19, abdominal obesity, and age are factors independently associated with antibody response” and finally the data on antibody levels induced by the vaccination schedules are presented.

9. It seemed better for us to explain what was done in the methods and in the results to only show what was observed, in order to not be repetitive.

10. It was defined as a positive association by the sign of the Beta value, which was positive, indicating that as the values of our independent variable increased, those of the dependent variable increased, and in the negative association, the beta value was negative, indicating that, when the value of the independent variable increase, the dependent variable decreased. For each sample, we use a different model, that is why when talking about days 180 and 365, we mention that they are different models than those of the previous samples.

11. It is specified in each part of the results whether it is a positive or negative association.

12. We present the data in sequential order:

Figure 1. Clinical profiles by (A) prior COVID-19, (B) COVID-19 post-vaccination in all population.

Table 2. Influence of abdominal obesity on the antibody response induced by vaccines in individuals without and with COVID-19 prior to vaccination

Figure 2. Clinical profiles by vaccination schedule in all population; (A) AZD1222, (B) Convidecia, (C) BNT162B2 (D) Sputnik V and (E) CoronaVac

Table 3. Analysis of the association of anthropometric factors with the anti-S1/S2 and anti-RBD IgG antibody neutralizing response in the study population

Finally, the figures on antibody levels induced by the vaccination schedules are presented.

13. We added more details about the experiments in all figure and table legends.

14. The word behavior was eliminated.

15. We replaced “RBD” with “Anti-S1/S2 and anti-RBD neutralizing IgG antibody” in title “3.2. Anti-S1/S2 and anti-RBD-neutralizing IgG antibody response induced by COVID-19 in AO+ individuals”

16. It was defined in its first use; IMSS (Mexican Social Security Institute), Participants Without Abdominal Obesity (AO-) or With Abdominal Obesity (AO+).

17. English correction was made. Certificate attached (Supplementary)

18. We previously reported the Association of obesity with SARS-CoV-2 and its relationship with the COVID-19-induced humoral response prior to vaccination. We observed a positive correlation in antibody levels with BMI and that obesity is an independent risk factor for SARS-CoV-2 infection when adjusted for confounding variables. For this reason, in the present study we evaluate the potential association between abdominal obesity and the levels of antibodies induced by COVID-19 and, more importantly, by 5 different vaccination schedules that were applied in the Mexican population, since it has been reported to the abdominal obesity as a more sensitive parameter of risk for diabetes, cardiovascular diseases and death than BMI, and has been associated with an increased risk of developing severe COVID-19. Additionally, we compared self-reported adverse events generated by vaccination and COVID-19 symptomatology in people with abdominal obesity with those without.

19. The corresponding statistics were made and we observed in the figure 6 legend: A) Median antibody levels in participants without history of COVID-19; participants in the AZD1222, Sputnik V and CoronaVac group without previous COVID-19 had significant differ-ences in their antibody levels between AO+ and AO- (21, 360; 21, 90; 21, 90 d, respectively). (B) With a history of prior COVID-19; participants in the AZD1222 and Convidecia group with previous COVID-19 had significant differences in their antibody levels between AO+ and AO- (0, 180; 360 d, respectively).

20. It is suggested that this could be the explanation for the high levels of antibodies in people with abdominal obesity, as we mentioned in the paragraph:

21. There are some studies where they observe higher levels of antibodies induced by COVID-19 in people with obesity, as we mentioned in the discussion:

22. The conclusion of the association of abdominal obesity with a worse prognosis when suffering from COVID-19 is relevant, since it is a vulnerable population as has been previously reported (higher risk of suffering from severe COVID-19, is related to comorbidities such as diabetes and cardiovascular diseases) and with this type of study, the importance of vaccination is emphasized in this population, in which a reduction in symptoms was also observed when suffering from COVID-19 after vaccination, complementing the results of the adverse events, where we observed that all the vaccination schedules evaluated are safe for the population of interest in this study. These results can help us improve vaccination schedules, giving priority to an increasing population in the world and especially in Mexico.

Reviewer 2 Report

Comments and Suggestions for Authors

The study submitted for review is of interest for daily clinical practice. The association of obesity with severe disease due to COVID was evident from the beginning of the pandemic, and there was little data to justify the reason for its occurrence. The relationship between the levels of antibodies produced by the different vaccines and the increase in body mass index has not been studied and the results obtained by the authors are of interest. 

The article is well structured and the results are derived from the questions posed.

I would accept the paper for submission in its present form. 

Author Response

Thanks for your review

Reviewer 3 Report

Comments and Suggestions for Authors

The authors are studying biostatistical relationsships between COVID vaccination and infection, measures of body mass and obesity, and outcomes such as antibody levels and

longevity. The study design is complicated and explained less clearly than it could be. One of the key points seems to be that Abdominal Obesity is a more

useful measure of obesity than is the commonly used body mass index.

The writing at sentence level is satisfactory; the problem is at a higher level of organization at which I found it hard to understand the logical and immunological flow of the manuscript.

Two of the potential strengths of this study are that:

a) the study is done in Latin America, whereas most studies have been done in Europe, Israel, USA, and Asia

b) a larger variety of vaccines were given in Mexico than for example, in Israel

but these potential strengths are not evident in the Introduction as it is currently written

Major concerns:

1. The results in Figure 5, suggest that neutralizing antibody levels are very high through teh sixth period of measurement which is one year

post-vaccination. This finding contradicts several published studies that showed substantial declines 6 months after vaccination (references 1-5 below)

Those studies formed the basis for the recommendation to give boosters in many countries. The unexpected results in Figure 5 need substantial explanation.

2. The results in Figures 3 and 4 and in Table 3 imply that obese individuals have much higher antibody levels and this explained in subsection 3.7, but if that is true, then why is obesity

associated with worse outcomes (as the authors delineated in the Introduction)?.

The authors recognize this unexpected outcome in the Discussion, but the text there is poorly organized.

They discuss the relationship between obesity and antibody levels in two separate paragraphs that start

"Regarding the levels of antibodies induced by COVID-19 (line 373) and "In our study, vaccine-induced antibody levels were higher in the WAO group (line 402)".

The entire text between these two paragraphs needs to be reorganized to make clear how the results in this study are consistent or inconsistent with related studies from

other countries. in the present writing, I simply cannot decide how the results compare to previous publications.

3. The findings in the subsection entitled "Correlation of IgG and RBD with participant age" are unclear to me. The subsection should be rewritten.

4. The Abstract is poorly written and lacks a clear delineation of the main results.

Minor concerns:

5. One should avoid use of acronyms in the Abstract

6. The following acronyms are not explained at the first usage: ENSANUT, CVD, IMSS

7. Line 24 and elsewhere, replace "Our country" with "Mexico"

8.  WAO, WoAo are not defined. Using AO+ and AO- would be clearer.

9. Lines 34,76 'reactogenicity" is not a standard term and needs to be defined.

10. Lines 47-57, The authors shoudl  do a more extensive literature search for studies relating obesit and COVID-19 and document in which countries

studies were done and which vaccines were given in those countries.

11. Lines 58-60, I do not understand the sentence that starts "On the other hand". The authors should split the sentence into multiple one-clause sentences.

12. Lines 113-116, why did the authors measure the waist circumference but rely on self report for weight and height?

13. Linw 190, It is not  clear what Figure 1 is showing because I do not undestand what the authors mean by "response rate"

14. Header of Table has "0.275" where I believe that the authors intended to put "Beta"

15. In the legend of Figure 1, change "Shakin" to "Shaking" [typo]

16. In Figure 2, the horizontal bar for the mean are not visible for several of of the WoAO plots

17. Line 490, the verb "relapse" does not make sense here; maybe the authors meant "lies"

References (to document decrease in antibodies six months after vaccination)

1. Israel A et al., Large-scale study of antibody titer decay following BNT162b2 mRNA Vaccine or SARS-CoV-2 infection. Vaccines 10(2021), article 64.

2. Naaber P, et al. Dynamics of antibody response to BNT162b2 vaccine after six months: a longitudinal prospective study. Lancet Regional Health Europe 10(2021), article 100208.

3. Sughayer MG et al., Comparison of the effectiveness and duration of anti-RBD SARS-CoV-2 IgG antibody response between different types of vaccines: Implications for vaccine strategies.

Vaccine 40(2022), 2841-2847.

4. Terpos E et al., Comparison of neutralizing antibody responses at 6 months post vaccination with BNT162b2 and AZD1222, Biomedicines 10(2022), article 338.

5. Varona JF et al., kinetics of anti-SARS-CoV-2 antibodies over time. Results of 10 month follow up in over 300 seropositive Health Care Workers. European Journal of Internal Medicine 89(2021), 97-103.

Comments on the Quality of English Language

The writing at sentence level is satisfactory; the problem is at a higher level of organization at which I found it hard to understand the logical and immunological flow of the manuscript. In other words, the writing problem with this manuscript is language-independent and would have existed had the authors written in Spanish rather than in English.

Author Response

Reviewer 3

1. The reason why the high levels were maintained for a year is due to the application of a booster for all vaccines that is explained in section 2.2:

A booster (third dose) was applied to the general population in January 2022; of the total number of participants, only 84.8% (330 participants) received this vaccine dose. The main vaccine administered was AZD1222 (57.9%), followed by Sputnik V (22.7%) and Moderna (16.7%).

And in the section 3.8:

Finally, we again observed an increase in the levels of IgG and anti-RBD-neutralizing antibodies after 270 days, which remained high until the last sample (365 d) because the booster was administered to the entire population approximately 240 d after the first dose was administered (Figure 6).

2. The discussion was restructured as follows:

Nonetheless, the underlying physiological explanation for the elevated levels of IgG and neutralizing antibodies at the post-convalescent stage remains unclear. It is important to go deeper into the study of the levels and role of antibodies against SARS-CoV-2 in people with obesity as it has been seen that obesity impairs immune function, causing chronic inflammation by increasing the number of B cells in visceral adipose tissue and producing autoreactive immunoglobulins [34] and central adiposity is associated with an increased proinflammatory fraction of IgG [35]. In addition, it has been observed that a good proportion of the antibodies produced in people with COVID-19 and obesity are autoimmune [7,36]. It has been suggested that the antibody response could be associated with secondary organ damage mediated by antibodies in addition to antiviral efficacy, which could explain why the participants in our study with higher BMI or WC had higher levels of antibodies and more severe symptoms when they had COVID-19.

In our study, vaccine-induced antibody levels were high in the AO+ group. When stratified by vaccination scheme, we observed a significant difference in anti-S1/S2 IgG antibody levels between groups that received AZD122 and CoronaVac and differences in anti-RBD-neutralizing antibodies levels between the AZD1222, Convidecia, Sputnik V, and CoronaVac groups. Interestingly, there are reports of lower antibody levels in people with high WC or abdominal obesity induced by BNT162b2 vaccines in healthcare workers in Italy [20,21] and by an inactivated vaccine in Chinese individu-als [19]; both studies included people without a history of COVID-19. With this in mind, we analyzed participants without prior COVID-19 and observed elevated levels and a high seropositivity rate in the AO + group 21 d after their first immunization. In contrast, a study on Chinese individuals observed low antibody seropositivity rates in patients with AO [19]. Interestingly, the relationship between AO and antibody levels continued independent of other variables according to our multiple linear regression analysis. However, the levels of anti-S1/S2 and anti-RBD-neutralizing IgG antibodies decreased at 180 d (sample prior to booster or 3th dose), especially in AO+ participants without previous COVID-19 vaccination with AZD1222 and with Covidencia, who had lower levels than the AO- group, in the case of BNT162b2 and Sputnik V was not as evident. Sheridan et al. found similar results, showing a correlation between BMI and elevated baseline IgG antibody levels; 12 months after vaccination, a higher BMI was associated with a greater decrease in influenza antibody levels in the USA [37]. Interestingly, a study in an Israeli population observed that people with a low BMI (<18.5) vaccinated with BNT162b2 had lower antibody levels than those with a high BMI [38]. In contrast, a Greek study found no significant difference in the decrease in antibody levels at 6 months among people with different BMI vaccinated with BNT162b2 and AZD1222 [39]. It is possible that the study ethnic group could also in-fluence the longevity of antibodies in people with obesity, as observed that among those vaccinated with BNT162b2, antibody titers were increased in Arab and Jewish Ultra-orthodox individuals compared with the general population [38].

Despite the difference in vaccination-induced antibody levels between the AO+ vs. AO-, in both groups the frequency of hospitalization and use of oxygen decreased, showing that the vaccines confer protection against the severity of COVID-19 in all participants, which is similar to other studies, where the efficacy of mRNA vaccines against SARS-CoV-2 does not differ between people with obesity compared to; that in people without obesity [40,41]. However, AO+ participants decreased their antibody levels in a shorter amount of time.

3. The section “Correlation of Anti-S1/S2 and anti-RBD-neutralizing IgG antibody with participant age was rewritten as follows:

Subsequently, we evaluated whether there was a correlation between the age of participants with prior documented COVID-19. We observed a correlation between age and high levels of antibodies in previous infections and with the application of the second dose of the vaccine (90 d) in the total population. When stratifying the participants by the presence or absence of AO, we observed that only the AO- group present-ed a correlation between age and anti S1/S2 IgG antibodies after the second dose (90 d). Interestingly, the levels of antibodies with neutralizing activity were negatively correlated with advanced age in the longest periods (270 and 365 d). When stratifying the population by documented prior COVID-19 infection, we observed something similar to that when stratified by the presence or absence of AO, and we observed a stronger positive correlation in shorter periods (0, 21, 90, and 180 d) in those who documented previous COVID-19 (Table S2).

4. The summary was rewritten as follows:

Abdominal obesity is highly prevalent in Mexico and has a poor prognosis in terms of the severity of coronavirus disease (COVID-19) and low levels of antibodies induced by infection and vaccination. We evaluated the humoral immune response induced by COVID-19 and five different vaccination schedules in Mexican individuals with abdominal obesity and the effects of other variables. This prospective longitudinal cohort study included 2,084 samples from 389 participants. The levels of anti-S1/S2 and anti-RBD IgG antibodies were measured at various time points after vaccination. A high prevalence of hospitalization and oxygen use was observed in individuals with abdominal obesity (AO) who had COVID-19 before vaccination; however, they also had high levels of an-ti-S1/S2 and anti-RBD-neutralizing IgG antibodies. The same was true for vaccination-induced antibody levels. However, their longevity was low. Interestingly, we did not observe significant differences in vaccine reactogenicity between abdominally and non-abdominally obese groups. Finally, individuals with a higher body mass index, older age, and previous COVID-19 had higher levels of antibodies induced by COVID-19 and vaccination. Therefore, it is important to evaluate other immunological and inflammatory factors to better understand the pathogenesis of COVID-19 in the presence of risk factors and propose effective vaccination schedules for vulnerable populations.

5. Acronyms were removed from the summary

6. National Health and Nutrition Survey; ENSANUT, IMSS; Mexican Social Security Institute

7. We replaced “our country” with “Mexico”

8. Without Abdominal Obesity; AO- or With Abdominal Obesity; AO+ (section 2.4)

9. We define reactogenicity as follows: reactogenicity (physical manifestations of the inflammatory response to vaccination: fever, tiredness, headache, muscle pain, shaking chills, diarrhea, and local reaction)

10. We added 6 more references (38-39, 45-48) and the countries where the reference studies were carried out were added (31-33)

11. The sentence was rewritten like this: However, BMI can be used only as an approximation of the degree of adiposity [3] because it does not consider body fat distribution, especially visceral fat. Visceral fat is a risk factor for several cardiometabolic diseases and is associated with a high mortality [10].

12. Waist circumference was measured because the project planned to evaluate antibody levels in people with and without abdominal obesity. In addition, it is an easy parameter to obtain, and a scale was not always available in the centers where we the samples and data were collected, making more difficult to obtain the weight and height values of the participants.

13. The sentence was rewritten like this: Each graph shows the proportion, from 0% (center of the circular graph) to 100% (circular graph perimeter).

14. We replaced “0.275” with “Beta” in Table 2

15. We replaced “Shakin” with “shaking” in figure 1 and 2

16. Figure 3 was edited

17. We replaced “relapse” with “lies”

Round 2

Reviewer 1 Report

Comments and Suggestions for Authors

The authors have revised the manuscript and it is clearer than the earlier version. 

Comments on the Quality of English Language

Minor English editing might be necessary. 

Author Response

Line 35, We changed "abdominally and non-abdominally obese" to "abdominally obese and abdominally non-obese"

Line 39, We changed "and propose" to "and to propose"

Line 118, We changed "main vaccine administered" to "main vaccine administered for the booster"

Line 144, We changed "Institute" to "institutes"

We changed “diarrohea” to “diarrhea” in figure 1

We changed “head ache” to “headache” in figure 1

Lines 214-215, We changed "all population" to "all populations"

Line 229, We changed "In contrast, the" to "the

Line 235, We changed "shows" to "showed"

Line 263, We changed "the data are presented as beta value" to "For each regression analysis, the results are presented in terms of the beta values."

Line 312, We changed "the same effect" to "comparably strong associations"

Line 347, We changed "difference in" to "difference between AO+ and AO- in"

Line 562, We changed "since the first" to "after the first"

Line 610, We changed "vaccines and provided" to "vaccines, which provided"

Reviewer 3 Report

Comments and Suggestions for Authors

The authors have revised the manuscript extensively and addressed all of my concerns with the original submission. There remain some minor errors in writing listed in the next box.

Comments on the Quality of English Language

Line 35, change "abdominally and non-abdominally obese" to "abdominally obese and abdominally non-obese" [grammar]

Line 39, change "and propose" to "and to propose" [grammar]

Line 79, define reactogenicity here rather than at line 118

Line 116, change "main vaccine administered" to "main vaccine administered for the booster" [clarification]

Line 144, change "Institute" to "institutes" [typo]

Lines 214-215, change "all population" to "all populations"

Line 229, change "In contrast, the" to "the [wrong meaning]

Line 235, change "shows" to "showed" [the rest of Results is appropriately written in past tense]

Line 263, change "the data are presented as beta value" to "For each regression analysis, the results are presented in terms of the beta values." [clarification]

Line 312, change "the same effect" to "comparably strong associations" [wrong meaning]

Line 347, change "difference in" to "difference between AO+ and AO- in" [clarification]

Line 538, delete blank line.

Line 562, change "since the first" to "after the first" [wrong word]

Line 610, change "vaccines and provided" to "vaccines, which provided" [wrong meaning]

Author Response

Line 35, We changed "abdominally and non-abdominally obese" to "abdominally obese and abdominally non-obese"

Line 39, We changed "and propose" to "and to propose"

Line 80, We defined reactogenicity here rather than at line 118

Line 118, We changed "main vaccine administered" to "main vaccine administered for the booster"

Line 144, We changed "Institute" to "institutes"

Lines 214-215, We changed "all population" to "all populations"

Line 229, We changed "In contrast, the" to "the

Line 235, We changed "shows" to "showed"

Line 263, We changed "the data are presented as beta value" to "For each regression analysis, the results are presented in terms of the beta values."

Line 312, We changed "the same effect" to "comparably strong associations"

Line 347, We changed "difference in" to "difference between AO+ and AO- in"

Line 538, We deleted blank line.

Line 562, We changed "since the first" to "after the first"

Line 610, We changed "vaccines and provided" to "vaccines, which provided"